# Lipin-1, a Versatile Regulator of Lipid Homeostasis, Is a Potential Target for Fighting Cancer

**DOI:** 10.3390/ijms22094419

**Published:** 2021-04-23

**Authors:** Laura Brohée, Julie Crémer, Alain Colige, Christophe Deroanne

**Affiliations:** Laboratory of Connective Tissues Biology, GIGA-Cancer, University of Liège, 4000 Liège, Belgium; laura.brohee@gmail.com (L.B.); Julie.Cremer@uliege.be (J.C.); acolige@uliege.be (A.C.)

**Keywords:** lipin-1, cancer, metabolism, lipids, propranolol, fatty acids, phosphatidic acid phosphatase

## Abstract

The rewiring of lipid metabolism is a major adaptation observed in cancer, and it is generally associated with the increased aggressiveness of cancer cells. Targeting lipid metabolism is therefore an appealing therapeutic strategy, but it requires a better understanding of the specific roles played by the main enzymes involved in lipid biosynthesis. Lipin-1 is a central regulator of lipid homeostasis, acting either as an enzyme or as a co-regulator of transcription. In spite of its important functions it is only recently that several groups have highlighted its role in cancer. Here, we will review the most recent research describing the role of lipin-1 in tumor progression when expressed by cancer cells or cells of the tumor microenvironment. The interest of its inhibition as an adjuvant therapy to amplify the effects of anti-cancer therapies will be also illustrated.

## 1. Introduction

Cancer cells frequently show altered metabolic activities compared to normal tissues [1]. This metabolic rewiring is essential for cancer cells to sustain their increased needs for energy and building molecules. It is also guided by the harsh tumor microenvironment, which appears unfavorable for cell growth and survival [2]. One of the best-known metabolic modification in cancers, commonly called aerobic glycolysis or the Warburg effect [3], is the increase in glucose uptake and its conversion into lactate. Despite being a less efficient energy source, aerobic glycolysis is essential for several cancers. Furthermore, lactate itself can play a role in the development and maintenance of the cancer [4,5]. In parallel, cancer cells undergo a full reprogramming of lipids metabolism, and become addicted to fatty acids and cholesterol uptake and/or synthesis to generate the new membranes and organelles required for cell growth, division and migration, which is essential for fast growing cells and for processes such as metastasis [6]. The re-orientation of cancer cells’ metabolism towards lipids biosynthesis is especially striking after cancer relapse, and it contributes to the cancer’s enhanced aggressiveness [7,8]. Therefore, it is not surprising that an increasing number of enzymes involved in lipid biosynthesis have been considered as privileged targets in the context of the development of anti-cancer therapies [9]. Targeting de novo fatty acid synthesis, for example, has shown efficacy in repressing proliferation and migration, and in inducing cell death in several cancer types without affecting normal cells [10,11,12]. However, the promising results obtained in preclinical models with drugs targeting lipid biosynthesis have not been successfully translated into the clinic [13], and the characterization of the role of other enzymes involved in lipid homeostasis is expected to lead to innovative strategies to overcome this current limitation. Among the least studied enzymes in lipid biosynthesis are those of the lipins family [14], also called phosphatidic acid phosphatases (PAPs), which includes three members (lipin-1, lipin-2 and lipin-3) in mammals [15]. Lipins play a pivotal role in lipid homeostasis, exerting a dual function as an enzyme involved in de novo lipid synthesis and also as a transcriptional co-regulator of fatty acids oxidation genes (Figure 1) [16]. The major lipid synthesis mechanism in cells is the glycerol-3-phosphate pathway [17]. Lipins catalyze the limiting step by de-phosphorylating phosphatidic acid into diacylglycerol [15,18], and therefore regulate the synthesis of several lipid species, including phospholipids, which are especially needed for membrane formation in fast-growing cells. Studies on intrinsic PAP activities have showed that lipin-1 has the highest kinetic value (V_max_), followed by lipin-2 and lipin-3 (V_max_L1 >> V_max_L2 > V_max_L3) [15]. Via its enzymatic activity, lipin-1 was reported to promote several processes, including cell differentiation, inflammation and autophagy, which can contribute to both cancer initiation and progression [19,20,21,22,23,24]. More recently, it was reported that extracellular mechanical signals, which are altered in several cancer types, regulate lipid homeostasis via the direct modulation of lipin-1 enzymatic activity [25].

In the nucleus, lipin-1 interacts with several transcriptional factors regulating the expression of genes involved in lipid homeostasis. Lipin-1 can increase PPARα and PGC1-α activity, while being itself a target gene of PGC1-α, which establishes a positive regulatory loop [26]. Via direct or indirect interactions, lipin-1 was also reported to positively regulate PPARγ, HNF-4 and C/EBPα, and to negatively regulate NFATc4 and SREBP1 [20,27,28,29,30].

In other species, lipin was also reported as a main regulator of lipids biosynthesis, affecting pathways that are relevant to the control of tumors in mammals. In Caenorhabditis elegans, lipin-1 is involved in lifespan control by maintaining polyunsaturated fatty acids expression [31], while in Drosophila, dlipin was notably reported to regulate the PI3K-Akt pathway as well as fatty acid beta-oxidation [32,33]. Hence, due to its central role in lipid homeostasis, it is not surprising that lipin-1 was recently identified as a regulator of cancer progression. General information regarding lipins’ structure, post-translational modification or general regulation have been recently reviewed [34,35], and will not be further described. In this review, we will focus on the most recent research describing the roles of lipin-1 in tumor progression, illustrating the interest in its inhibition to supplement other anti-cancer therapies.

## 2. Lipin-1 Is Overexpressed in Several Cancer Types and Regulates Cancer Cells Phenotype

The first clue that lipin-1 could be important for cancer progression was reported by Brohee et al. in a transcriptomic analysis on human prostate adenocarcinoma cells (PC3), showing that the expression of lipin-1 is regulated by the small RhoGTPase Rac1 [36]. Since Rac1 is often overexpressed and/or overactivated in cancers, the hypothesis that lipin-1 may be involved in tumor biology was raised. The implication of lipin-1 in cancer biology was also suggested by its overexpression in several cancer cell lines and in high-grade prostate adenocarcinomas. A correlation with cancer aggressiveness was also confirmed in models showing that lipin-1 silencing increases autophagy, and strongly represses prostate and breast cancer cell proliferation and migration without affecting normal cells. The knockdown of lipin-1 or the inhibition of its PAP activity by propranolol sensitized breast and prostate cancer cells to the inhibition of mTORC1 by rapamycin, hence providing experimental evidence that lipin-1 represents a potential target for cancer treatments [36].

The overexpression of lipin-1 was confirmed in situ in prostate cancer and in other cancer types, notably in triple negative breast cancer (TNBC) and in lung adenocarcinomas, where it is associated with poor prognosis [37,38,39]. Lipin-1 is also part of a gene expression signature for TNBC in a study comparing TNBC to non-TNBC tumors [40]. Recently, a negative correlation has been established between the expressions of lipin-1 and of the p53 tumor suppressor gene, suggesting a role for p53 in the transcriptional regulation of lipin-1 during cancer progression [41].

## 3. Mechanisms Regulated by Lipin-1 in Cancer

Several intracellular pathways regulated by lipin-1 are involved in cancer cell biology (Figure 2). He and collaborators observed that lipin-1 knockdown induces the apoptosis of TNBC cells, while barely affecting normal control cells and estrogen-dependent cell lines. Since lipin-1 is a phosphatidic acid phosphatase and catalyzes the rate-limiting step in phospholipid biosynthesis, its knockdown in TNBC cell lines affects the production of several phospholipids. This unbalanced production of lipid species induces endoplasmic reticulum (ER) stress, leading to the activation of the IRE1–XBP1 unfolded protein response pathway. They also demonstrated that lipin-1 knockdown reduces the growth of tumor xenograft [38]. Similarly, Fan and collaborators observed that lipin-1 silencing decreases proliferation and increases apoptosis in lung adenocarcinoma (LUAD) cell lines, while barely affecting normal control cells. In this model, lipin-1 knockdown increases the initiation of autophagy, as demonstrated by the enhanced phosphorylation of ULK1 (Unc-51 autophagy activating kinase 1) at Ser555 and of beclin-1. It also impairs autophagy clearance by decreasing the fusion between autophagosomes and lysosomes. Confirming the observation made by He et al. in TNBC cells, Fan and colleagues noticed enhanced ER stress upon lipin-1 silencing in LUAD cell lines. They identified the mechanism behind ER stress as an activation of the IRE1alpha–JNK pathway, which triggered the autophagy dysregulation. Indeed, the inhibition of this pathway partly restored autophagy and rescued the inhibition of proliferation and the induction of apoptosis observed upon lipin-1 silencing [37]. These regulations are attributable to a lack of products generated by the enzymatic function of lipin-1. In the same line of thought, Brohee et al. reported that lipin-1 depletion inhibits prostate cancer cells migration through an upregulation of RhoA activity, likely due to the increased phosphatidic acid concentration resulting from lipin-1 silencing [36]. However, lipin-1 also regulates cancer cell phenotype through other signaling pathways. In 2016, Kim et al. reported that the overexpression of lipin-1 promotes epithelial transformation and tumorigenesis in breast cancer cells. They observed that lipin-1 expression was correlated with the level of insulin-receptor substrate 1 (IRS1) in breast tumor samples and that lipin-1 co-localizes and directly interacts with IRS1. Lipin-1 contributes to increase IRS1 protein levels by inhibiting its degradation, thus enhancing the RAF1–MEK–ERK–p90RSK pathway. Kim et al. provided evidence that the activation of this axis induces c-fos expression and subsequent activator protein-1 (AP-1) activation, leading to enhanced tumor growth [42].

## 4. miRNAs as Tumor Suppressors Targeting Lipin-1

miRNAs are small non-coding RNAs that can regulate the main cellular functions, including metabolism, by interacting with target mRNAs. Zhao and collaborators observed that mir451a can act as a tumor suppressor on hepatocellular carcinoma (HCC). Via database analyses, they identified lipin-1 mRNA as a target of mir451a, and demonstrated that mir451a exerts its anti-tumoral action by silencing lipin-1 in endothelial and HCC cells [43]. Similarly, Yang and Ma reported that mir584 regulates ovarian cancer progression by targeting lipin-1. In ovarian cancer, the expressions of mir584 and of lipin-1 are regulated in opposite directions, and depleting the cells of mir584 induces an increase in lipin-1 expression, resulting in increased tumorigenesis. Furthermore, they also showed that low mir584 levels correlate with increased metastasis spreading and poor prognosis [44].

## 5. c-src and mTORC1 Target Lipin-1 to Regulate Cancer Progression

Post-translational modifications of lipin-1 are involved in the regulation of its PAP and transcriptional cofactor activities, and have been reviewed elsewhere [35]. However, recently identified post-translational modifications seem to play an essential role in cancer. c-src is a proto-oncogene regulating various cellular functions, including proliferation, migration, lipogenesis and apoptosis. Song et al. recently demonstrated that c-src can phosphorylate lipin-1 on tyrosine 795, which increases its phosphatidic acid phosphatase activity and, therefore, the synthesis of triacylglycerol and several phospholipids. They demonstrated also that the c-src-lipin-1 axis and the phosphorylation of lipin-1 on tyrosine 795 are essential in breast cancer cells for their proliferation in vitro, and for the formation of tumors in vivo. Moreover, the analysis of human samples demonstrated that this c–src–lipin1 axis is constitutively hyper-activated in human breast tumors as compared to adjacent normal tissue, and that the phosphorylation of lipin-1 is associated with a poorer prognosis [45].

mTORC1 is a master regulator of growth and, a hyper-activation of its signaling pathway has been correlated with poor prognosis for many cancer types. It is also important to underline that mTORC1 can phosphorylate lipin-1 on multiple sites, and that these sequential phosphorylations target lipin-1 to the ER membrane, which, due to the proximity to its substrate, enhances its PAP activity and promotes cancer growth [29]. The modulations of lipin-1 activity mediated by mTORC1 can affect mitochondria dynamics and regulate cancer. The metabolic flexibility required for cancer cells to adapt to their changing microenvironment is provided by the reprogramming of mitochondria. The metabolic function of mitochondria is coupled to fusion and fission events, which shape their morphology. MacVicar et al. [46] demonstrated that the protease YME1L, which reshapes mitochondria, is affected by lipin-1 enzymatic activity, itself under the control of mTORC1. They observed that this mTORC1–lipin-1–YME1L-dependent reshaping of mitochondria supports pancreatic ductal adenocarcinoma development.

These studies support the concept that lipin-1 is a central node involved in the main intracellular signaling pathways regulating cancer cells’ metabolism.

## 6. The Role of Host Lipin-1 in Inflammation-Driven Cancer Progression

Lipin-1 is an essential protein for macrophages’ activity, because they rely on a tight regulation of lipid catabolism and anabolism to fully exert their function and to effectively contribute to tissues restoration. The co-transcriptional function of lipin-1 is required in these cells to keep this balance and, upon lipin-1 silencing, macrophage activity is reduced [47].

Chronic inflammation is the basis of the initiation and the progression of several cancer types, including colon cancer. The use of anti-inflammatory treatments can reduce the risk of cancer development and can improve conventional therapies’ outcomes [48,49]. In this context, lipin-1 can regulate, dependently and independently of its enzymatic activity, the response of macrophages to pro-inflammatory stimuli. In animal models displaying pro-tumorigenic systemic inflammation, lipin-1 deficiency decreases the damage due to the overexpression of pro-inflammatory molecules [21,22,50]. Meana and colleagues explored the potential role of host lipin-1 in the development of colon cancer. They elegantly demonstrated that, in a model of colitis-associated tumors, lipin-1-deficient animals developed fewer and smaller tumors than their littermate controls. They also observed the decreased M1-like activation and infiltration of macrophages in tumors of lipin-1-deficient mice. The graft of WT macrophages into lipin-1-deficient mice, or their treatment with the pro-inflammatory interleukin 23, increased colitis burden. Finally, the analysis of several clinical datasets confirmed that high lipin-1 expression is correlated with pro-inflammatory cytokines’ expression and with poorer prognosis in two subtypes of colon cancer [51]. However, it must be highlighted that lipin-1 not only contributes to pro-inflammatory macrophage responses, but also participates in their pro-resolving function, notably via its transcriptional coregulatory activity [47]. This suggests that the selective targeting of individual lipin-1 activities may have better outcomes than simply altering total levels of lipin-1.

Taken together, these studies clearly show the pivotal role that lipin-1 can play in host cells to generate a favorable environment for cancer establishment and progression. Further studies are, however, required in order to fully understand how host lipin-1, but also host lipin-2 and -3, can alter the tumor microenvironment.

## 7. Lipin-1 Silencing as a Novel Adjuvant Therapy for Cancer Treatment

An ever-increasing number of studies aim at identifying new combinatory treatments to improve cancer therapy and override acquired resistances leading to cancer relapse. For example, much effort has been devoted to inhibiting the PI3K/AKT/mTOR pathway, which is overactivated in various cancer types. The inhibition of mTORC1 by rapamycin was successful in some malignancies [52,53], but, in numerous cancer types, it has only limited effects due to the loss of negative feedback loops that lead to the reactivation of AKT by mTORC2 [54]. Since lipin-1 inhibition in cancer cells leads to the reduction of AKT and S6 protein activation, the logical next step was to combine rapamycin treatment with either lipin-1 silencing or its pharmacological inhibition. Both strategies were found to synergize with rapamycin treatment in inhibiting prostate and breast cancer cells’ proliferation, confirming the hypothesis that lipin-1 targeting can be used as an adjuvant therapy [36].

Another approach consists in targeting PI3K, yet the cytotoxicity towards cancer cells of specific PI3K inhibitors is relatively weak. In order to enhance its efficacy, Saijo and colleagues developed a new dual histone deacetylase (HDAC)/PI3K inhibitor (FK-A11), but, again, with little success in vivo [55]. In order to identify an innovative adjuvant therapy to be used with their dual inhibitor, in a follow-up study, Imai and colleagues performed high-throughput screening. Using this approach, they found that lipin-1 inhibition amplifies the cytotoxicity of FK-A11, both in vitro and in vivo [56].

Autophagy is known to be induced upon chemotherapy, and is generally viewed as contributing to resistance to treatments [57]. For example, cells can evade the cytotoxic effect of drugs, such as cisplatin, by activating autophagy. Fan and collaborators observed that, in lung adenocarcinoma cells, lipin-1 silencing increases autophagy initiation, but disrupts the formation of functional autolysosomes. Therefore, they combined lipin-1 inhibition with cisplatin treatment and observed that lipin-1 silencing sensitizes these cancer cells to cisplatin treatment, most likely by disrupting the autophagy flux [37].

Very recently, it was also reported that CZ415, a novel pharmacological inhibitor of mTOR, which has potent anti-tumor properties in several types of cancer, mediates its effects in cervical cancer models via lipin-1 inhibition [58]. Although further studies are required to better elucidate how CZ415 inhibits lipin-1 activity via the mTOR axis, this study is a further confirmation of the high potential of lipin-1 targeting in cancer therapies alone or in combination with already established treatments.

## 8. Propranolol, a Pharmacological Inhibitor of Lipin Enzymatic Activity, Alters Autophagy Homeostasis

Propranolol is one of the most studied repurposing candidates in oncology. First developed as a β-adrenergic inhibitor [59,60], it was shown later that it is a potent inhibitor of the PAP activity of lipins [61,62]. Propranolol is a drug widely used in clinic for a variety of diseases, and it has minimal toxicity. It has been evaluated alone or in combination for the treatment of several cancers, including breast and prostate cancers [63,64]. In combined treatment, it can amplify the cytotoxic effect of chemotherapeutic agents like vinblastine, doxorubicin, cisplatin and metformin [65,66,67,68]. By inhibiting the PAP activity of lipins, propranolol blocks the fusion between lysosome and autophagosome, which leads to the massive accumulation of immature autophagosomes and to an overall blockage of autophagy [69]. The therapeutic effects of some drugs are limited by the induction of survival autophagy that protects cancer cells from apoptosis and death. When propranolol is combined with such drugs, such as the glycolysis inhibitor 2-deoxy-d-glucose, or with starvation conditions, it induces a dramatic accumulation of early autophagosomes. This blockage of autophagy leads to the aggravation of ER stress, a mechanism widely exploited and very efficient in cancer therapy [70]. Brohee et al. showed that these multiple stresses cause a metabolic catastrophe in the cancer cells, which compromises their adaptation capacities, and ultimately leads to cancer cell death in vitro and the inhibition of tumor growth in vivo [71]. Similar results were reported by Lucido and collaborators by combining propranolol with dichloroacetate, another inhibitor of glycolysis [72]. In addition, the disruption of autophagy by propranolol was also reported to amplify the cytotoxic effect of cisplatin in lung cancer models [37], highlighting the potential of propranolol as an interesting alternative to other autophagy inhibitors, which are relatively toxic in noncancerous tissues as well [73]. On this basis, it is clear that propranolol is an extremely promising candidate for defining novel combined therapies in cancer treatments with little to no effect on normal tissues.

## 9. Conclusions and Perspectives

The recent studies reviewed here have highlighted the significant role of lipin-1 in cancer cell phenotype and the interest of its inhibition for anticancer therapy, notably because it efficiently targets several cancer cell types without affecting normal control cells [36,37,38,42]. It is still important, however, to better understand its various contributions to cancer progression. Lipin-1 has a dual function as an enzyme involved in the Kennedy pathway, but also as a transcriptional cofactor regulating the expressions of genes involved in lipid homeostasis. It is therefore of prime importance to define the respective roles of these two functions, which have not been thoroughly investigated up to now. Another aspect of lipin-1 that is still understudied concerns its variants resulting from an alternative splicing mechanism. The lipin-1 pre-mRNA can undergo alternative splicing, potentially generating three variants, with lipin-1α and lipin-1β being the most commonly expressed [74]. These lipin-1 variants are very similar, and only differ by a few amino acid insertions or deletions [74]. Although being very similar, these variants display different PAP activities and can specifically regulate the expression of different groups of genes [74,75]. Their expression pattern in tissues is also different, probably fulfilling different needs in different tissues [76], and the alteration of the ratio between the lipin-1 variants can contribute to pathogenesis, such as alcoholic fatty liver disease [77]. Thus far, studies investigating the role of lipin-1 in cancers have considered lipin-1 as a whole, without considering the respective roles of its different isoforms. Further studies are needed to untangle the specific role of each variant in tumor progression. It should be informative to analyze the relative expressions of lipin-1 variants, notably in breast, prostate and lung cancers, whose progression is dependent on lipin-1, to identify which variant(s) should be preferentially targeted to repress tumor progression. The ratio between several lipin-1 variants could be important for cancer progression, and can be modulated through the targeting of splicing factors, such as SRSF10, which regulates the ratio of lipin1α/lipin1β [77,78].

To complete our knowledge of the roles of lipins in cancer progression, it is also necessary to extend our investigations to lipin-2 and lipin-3, since no study has been specifically designed to address their roles in cancer. Indeed, their potential implication cannot be neglected, since they can mitigate the effect of silencing lipin-1 by compensating for its PAP activity. For instance, it was reported that the silencing of lipin-1 induces a compensatory upregulation of lipin-2 in prostate cancer cells, attenuating the effects of lipin-1 inhibition [36]. It is known that the regulation of the expressions of lipin-1 and -2 are quite specific. For example, PGC-1α glucocorticoids and pro-inflammatory cytokines regulate lipin-1 expression, but have no effect on lipin-2 [79]. As with lipin-1, lipin-2 also undergoes tissue-specific regulations. As an example, fasting induces lipin-2 overexpression in hepatocytes, but not in adipocytes [27,80]. Cellular stresses, such as ER stress, can also induce lipin-2 overexpression, specifically in hepatocytes [81]. Hence, the role of lipin-2 may be more important than that of lipin-1 in malignant tumors originating in tissues, where it is most expressed. Knowledge about the regulation of lipin-3 in cancer remains too limited to anticipate its potential roles, and therefore would deserve better characterization. Finally, as an alternative to siRNAs, which are potent investigation tools but still very difficult to translate into the clinics, pharmacological inhibitors could be used. Propranolol efficiently inhibits the phosphatidic acid phosphatase activity of lipins; however, it is not specific to lipins. Bromoenol lactone has been also successfully used to inhibit the enzymatic activity of lipins, but it is also not specific of lipins, as it targets Ca^2+^-independent phospholipase A2 with similar potency [82]. In this context, the recent characterization of the structure of lipin-1 [83] could potentially lead to the development of new inhibitors able to selectively target either the enzymatic or the co-transcriptional activities of lipins. Such an advance would constitute a milestone for translating experimental knowledge about lipin-1 into clinical perspectives.

## Figures and Tables

**Figure 1 ijms-22-04419-f001:**
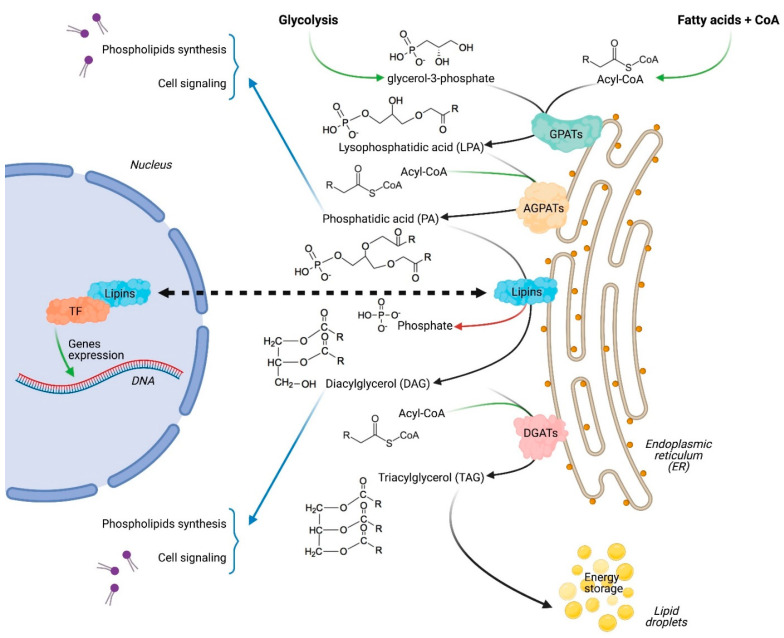
The dual function of lipins as enzymes in the glycerol-3-phosphate pathway on the endoplasmic reticulum and as transcriptional regulators in the nucleus by interacting with several transcription factors (TF).

**Figure 2 ijms-22-04419-f002:**
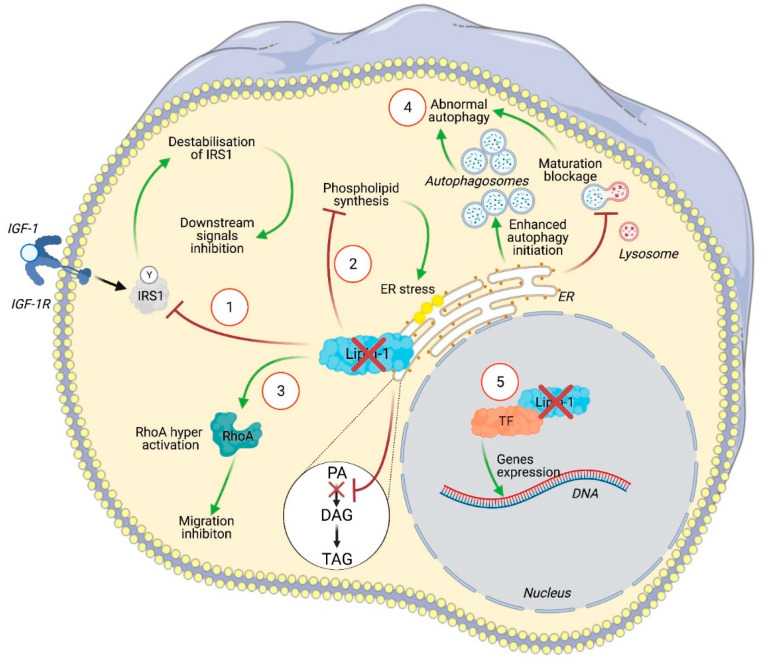
Illustration of the intracellular signaling pathways affected by lipin-1 silencing in cancer cells. In (1), insulin receptor substrate 1 (IRS1) is no longer stabilized by lipin-1 leading to an inhibition of Raf-1 and of its downstream targets including c-fos and AP-1. Lipin-1 silencing altered phospholipids synthesis (2). This activates the unfolded protein response (UPR) pathway and leads to endoplasmic reticulum (ER) stress. The increased concentration of phosphatidic acid (PA) due to the lack of lipin-1 activity can activate RhoA and inhibits cell migration (3). Lipin-1 depletion enhanced autophagy initiation, as indicated by enhanced LC3 and ULK phosphorylation, while a blockage of autophagosome maturation was observed, likely related to a lack of diacylglycerol (DAG). This leads to autophagosome accumulation (4). The precise contribution of lipin-1 transcriptional regulator activity to cancer cell phenotype has not yet been specifically addressed (5).

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
