# Peer review of "Lipin-1, a Versatile Regulator of Lipid Homeostasis, Is a Potential Target for Fighting Cancer"

_ijms, 2021, doi:10.3390/ijms22094419_

Round 1

Reviewer 1 Report

This is an excellent review that nicely sums up much of the new information of lipin-1 related to cancer biology. 2 areas that could be slightly improved. 1. A small addition on the contribution of lipin-1 to not only pro-inflammatory macrophage responses but also pro-resolving function I feel is warranted. Which actually suggests that selective targeting of individual lipin-1 activities may have better outcomes then simply altering total levels of lipin-1. 2. In the final paragraph I think mentioning that the characterization of lipin-1 structure (10.1038/s41467-020-15124-z) will likely lead to the identification and development on new small molecule inhibitors that can more selectively inhibit lipin enzymatic actiivty.

Author Response

Answers to Reviewer 1

This is an excellent review that nicely sums up much of the new information of lipin-1 related to cancer biology.

Answer: We thank the reviewer for his positive comment on our manuscript

2 areas that could be slightly improved.

  1. A small addition on the contribution of lipin-1 to not only pro-inflammatory macrophage responses but also pro-resolving function I feel is warranted. Which actually suggests that selective targeting of individual lipin-1 activities may have better outcomes then simply altering total levels of lipin-1.

Answer: A small paragraph mentioning the contribution of lipin-1 to the pro-resolving function of macrophages has been added (page 6, lanes 22-26)

  1. In the final paragraph I think mentioning that the characterization of lipin-1 structure (10.1038/s41467-020-15124-z) will likely lead to the identification and development on new small molecule inhibitors that can more selectively inhibit lipin enzymatic actiivty.

Answer: A small paragraph has been added according to the suggestion of the reviewer (page 8, lanes 35-38

Reviewer 2 Report

I found the review, Lipin-1, a versatile regulator of lipid homeostasis, is a potential target for fighting cancer by Laura Brohée , Julie Crémer , Alain Colige , Christophe Deroanne very compelling and update. The authors the role of the enzyme Lipin-in the  regulation of lipid homeostasis with the focus on its implication in cancer. I found this work interesting since Lipin-1 it may be a novel target for lipid metabolism in tumors, perhaps the only caveat is that I would have like to see a short paragraph on the important role of Lipin-1 in other species like Drosophila or C elegans where its role has been described to pathway relevant to the control of tumors in mammals, such as wingless, or in metabolic pathways that are linked to ageing. This is my only suggestion which perhaps maybe a little out the focus of the review but I think relevant to describe the role of Lipin and also for the readers-target of this review.

In conclusion I that this review is up to the standards of the journal and therefore I recommend it for publication, and if the authors feels to add something about the function of Lipin in other animal-models will be a plus.

Author Response

Answer to the comments of Reviewer 2

I found the review, Lipin-1, a versatile regulator of lipid homeostasis, is a potential target for fighting cancer by Laura Brohée , Julie Crémer , Alain Colige , Christophe Deroanne very compelling and update. The authors the role of the enzyme Lipin-in the regulation of lipid homeostasis with the focus on its implication in cancer. I found this work interesting since Lipin-1 it may be a novel target for lipid metabolism in tumors, perhaps the only caveat is that I would have like to see a short paragraph on the important role of Lipin-1 in other species like Drosophila or C elegans where its role has been described to pathway relevant to the control of tumors in mammals, such as wingless, or in metabolic pathways that are linked to ageing. This is my only suggestion which perhaps maybe a little out the focus of the review but I think relevant to describe the role of Lipin and also for the readers-target of this review.

In conclusion I that this review is up to the standards of the journal and therefore I recommend it for publication, and if the authors feels to add something about the function of Lipin in other animal-models will be a plus.

Answer: We thank the reviewer for his positive comments on our manuscript. A small paragraph underlining the important role of Lipin-1 in C. elegans and Drosophila has been added (page 3, lane 5-9)